# Best Practices to Support the Self-Determination of Indigenous Communities, Collectives, and Organizations in Health Research through a Provincial Health Research Network Environment in British Columbia, Canada

**DOI:** 10.3390/ijerph20156523

**Published:** 2023-08-04

**Authors:** Tara Erb, Krista Stelkia

**Affiliations:** 1School of Public Health and Social Policy, University of Victoria, Victoria, BC V8P 5C2, Canada; 2Faculty of Health Sciences, Simon Fraser University, Burnaby, BC V5A 1S6, Canada; krista_stelkia@sfu.ca

**Keywords:** Indigenous communities, collectives, and organizations (ICCOs), Indigenous community-based research, Indigenous health and wellness, self-determination, health research networks, Indigenous health research, Indigenous mentorship

## Abstract

In Canada, the health research funding landscape limits the self-determination of Indigenous peoples in multiple ways, including institutional eligibility, priority setting, and institutional structures that deprioritize Indigenous knowledges. However, Indigenous-led research networks represent a promising approach to transforming the funding landscape to better support the self-determination of Indigenous peoples in health research. The British Columbia Network Environment for Indigenous Health Research (BC NEIHR) is one of nine Indigenous-led networks across Canada that supports research leadership among Indigenous (First Nations, Métis, and Inuit) communities, collectives, and organizations (ICCOs). In this paper, we share three best practices to support the self-determination of ICCOs in health research based on three years of operating the BC NEIHR: (1) creating capacity-bridging initiatives to overcome funding barriers; (2) building relational research relationships with ICCOs (“people on the ground”); and (3) establishing a network of partnerships and collaborations to support ICCO self-determination. Supporting the self-determination of ICCOs and enabling them to lead their own health research is a critical pathway toward transforming the way Indigenous health research is funded and conducted in Canada.

## 1. Introduction

In Canada, the current health research funding landscape has limited and continues to limit the self-determination of Indigenous (First Nations, Métis, and Inuit) communities, collectives, and organizations (ICCOs) in conducting research that is grounded in their ways of knowing and being [1]. There are several barriers and challenges that have hindered this self-determination, including rigid institutional eligibility requirements for holding research funds from major funders, application requirements, and institutional structures that deprioritize Indigenous knowledges [2]. For example, Indigenous communities are not eligible to apply as the nominated principal applicant for federal research funds from the Canadian Institute for Health Research (CIHR) [3]. Currently, given that institution affiliation is a requirement of funding, the exclusion of ICCOs is seen as a consistent barrier to growing their capacity. While it is critical that ICCOs have ownership and control over research in which they are involved, they must also be supported in this leadership role. Supporting Indigenous health research leadership enables ICCOs to focus on exploring and addressing their own health priorities within the context of their distinct cultural, social, political, and economic environments. Therefore, capacity-bridging initiatives must recognize, mobilize, and support the capacities and strengths that exist within ICCOs [4].

Consideration of the historical context of research is essential to understanding the current landscape of Indigenous health research, and to thinking critically about the nature of research relationships [5]. Historically, research about Indigenous peoples was extractive in nature, inadequate in addressing the health and wellness of Indigenous communities, and often perpetuated health inequities, marginalization, and harm [5,6]. Furthermore, Indigenous knowledge-based research has been devalued within the Western scientific paradigm, where notions of “acceptable” research standards have led to it being criticized as “less valid” [7,8,9]. This is part of an ongoing systemic barrier that deprioritizes or dismisses Indigenous methods, methodologies, epistemologies, knowledge, and perspectives in health research [10]. In addition, other research has shown that systemic racism has negatively impacted and continues to negatively impact Indigenous participation and leadership in health research [11]. As a result, Indigenous communities across Canada report that transformations in Indigenous health research are needed [12,13], and, in response, Indigenous-led research networks have been created to reshape the funding landscape [14,15,16,17].

### The Need for Indigenous Leadership, Self-Determination, and Governance in Health Research

There is a critical need within Indigenous communities to address the historical impacts of unethical research by focusing on self-determination and supporting Indigenous ways of knowing in research. The harm caused by research and the unethical use of Indigenous data stems from a historical lack of power, which has prevented Indigenous communities from controlling and governing research processes and data, and this problem has only been amplified by the digital data revolution and the speed of technological innovation [18]. However, the socio-political structures and policies regarding the ethical governance of research are changing in response to Indigenous peoples becoming more self-determining [19]. Today, the principles of Ownership, Control, Access, and Possession (OCAP^®^), the Tri-Council Policy Statement 2 (TCPS2) Chapter 9 policy, the United Nations Declaration of the Rights of Indigenous Peoples Act (UNDRIP Act), the Declaration of the Rights of Indigenous Peoples Act (DRIPA), the emergence of distinct ICCO-led ethical guidelines, and the international forum Global Indigenous Data Alliance (GIDA) all have significant implications for research involving Indigenous people. The implications of such socio-political structures, policies, and legislation will facilitate the mobilization of ICCOs, enabling them to exert a high level of control and governance over research activities, support for the development of ICCO autonomous self-determined research programs for capacity building and knowledge transfer, and the asserting of Indigenous peoples’ jurisdiction and ownership over all data related to such research (i.e., data sovereignty) [20]. Most significantly, policy and legislation have the potential to provide Indigenous research participants with the means to push back and protect certain types of knowledge [21], which further promotes self-determination.

Increasingly, researchers who want to work with ICCOs are expected to demonstrate new forms of decolonial engagement that uphold Indigenous governance and self-determination and allow ICCOs to make final decisions over what knowledge will be shared, when, by whom, and in what ways [22]. In fact, when developing and defining research practices and projects related to Indigenous communities, the most important and contentious concerns regarding Indigenous research ethics for ICCOs are the issues of control, decision making, and governance (self-determination) [14,19,20,23,24]. Research that benefits and impacts Indigenous health and wellness is highly correlated to rethinking what constitutes ethical research and enacting the principles of Indigenous ownership and support for Indigenous governance and self-determination [25].

Indigenous and decolonial methodologies and Indigenous research ethics work together to uphold and advance Indigenous governance and self-determination. Indigenous methodologies include the unique ways researchers use Indigenous positionality and perspectives (a distinction-based approach) to conduct research with and within ICCOs [26]. New decolonial research methods created by Indigenous researchers and other oppressed groups promote a critical, reflexive view that shifts the research paradigm from Western hegemony to a decolonized agenda that centers on engagement and self-determination [27]. Decolonial methods honor governance and ICCO worldviews and knowledge systems when conducting research with ICCOs [28]. Decolonization is not about a total rejection of institutional processes or all that is Western or non-Indigenous; it is about a strategic agenda that focuses on the goals of governance and self-determination [29].

Ethical tensions around protecting Indigenous knowledge are ultimately about what counts as knowledge [23]; Indigenous ways of knowing are still not broadly accepted, respected, or protected within the institution or alongside Western knowledge. The struggle for Indigenous self-representation, a feature of self-determination, is a way to ensure that our cultural knowledge and resources will not languish in Western scholarship [20]. While many methodological and ethical challenges have yet to be addressed, efforts to negotiate ethical conditions in research aim to achieve the protection of Indigenous knowledges and control over Indigenous culture [24]. Indigenous peoples have the right to construct knowledge in the research environment in accordance with self-determined definitions of what is considered valuable and ethical [24], and this is a key principle of Indigenous health research networks. In this way, these networks hope to ethically invite and support ICCOs within the health research environment and to address and reduce the ethical tensions around protecting Indigenous knowledge.

In 2018, the CIHR announced the relaunching of the Network Environments for Indigenous Health Research (NEIHR) Program to provide supportive research environments for Indigenous health research driven by and grounded in Indigenous communities in Canada [30]. The British Columbia Network Environment for Indigenous Health Research (BC NEIHR) is one of nine such Indigenous-led NEIHR networks across Canada that supports research leadership among ICCOs. The BC NEIHR program was set up to purposively address the gap in institutional eligibility for ICCOs by having ICCOs—not academics—hold and control health research funds. In addition, through its leadership structure, the BC NEIHR addresses the gap in institutional structures that deprioritize Indigenous knowledges by working in partnership with funders, REBs, universities, health authorities, and many others to address anti-Indigenous racism in all research processes. The objective of this paper is to share three best practices to support the self-determination of ICCOs in health research based on the first three years of operating the BC NEIHR.

## 2. Methods

### 2.1. Overview of the BC NEIHR

The BC NEIHR (British Columbia Network Environment for Indigenous Health Research) is one of nine such NEIHR centers that support research leadership among ICCOs, Indigenous and non-Indigenous researchers, and Indigenous trainees in BC. The overarching goal of the BC NEIHR is to contribute to the improved health, wellbeing, and strength of Indigenous peoples by creating strong and respectful research relationships and supporting transformative research. We aim to transform research practices by supporting Indigenous-led health research in BC in the following ways: (1) develop infrastructure that supports ICCO-based health research; (2) facilitate and support Indigenous peoples in leading and controlling health research that reflects their values, priorities, and approaches; (3) provide funding to support ICCOs, including research development funds and knowledge-sharing and mobilization funds; (4) facilitate and support ethical and culturally safe research partnerships; and (5) engage policy and organizational partners at local, regional, national, and international levels to advance these objectives and ensure the sustainability of the BC NEIHR.

### 2.2. Operations Team

With guidance from the Governing Council, the Operations Team is responsible for the implementation of BC NEIHR policies, programs, and activities. The Operations Team includes Dr. Jeffrey Reading (Nominated Principal Investigator), knowledge users from First Nations Health Authority, Indigenous faculty from four universities across BC, and the NEIHR Network Coordinator. With support from the Indigenous Health Research Facilitators and partners, the Operations Team engages in collaborations to fulfill the following responsibilities: policy implementation, program development and implementation, and partnership and funding development.

### 2.3. Governance Strucutre

The BC NEIHR is governed by an Indigenous Governing Council that consists of nine members, including three Elders/Knowledge Holders, and representation from each of our three key community-based partners (the First Nations Health Authority, Métis Nations BC, and BC Association of Aboriginal Friendship Centers) as well as Indigenous student representation. Not only does this ensure that the BC NEIHR is led by Indigenous values and practices, but it ensures that Indigenous health research leaders are represented at the national NEIHR table, where they have a voice with which they can influence national and international Indigenous health research policies and programs. The governing council meets three times a year and engages in ongoing email communication. Through a consensus-based decision-making process, the Governing Council is tasked with advising the BC NEIHR and its team on key research priorities and strategic policies related primarily to: (a) network membership; (b) funding programs; (c) capacity-bridging/strengthening programs; (d) partnerships; and (e) future funding.

### 2.4. BC NEIHR Network Membership

The BC NEIHR is a large provincial Indigenous health research network that is dynamic and constantly changing over time. As of June 2023, we have over 300 registered members, including Indigenous trainees (37.8%), allies (22.4%), Indigenous academics/professionals (21.4%), and ICCOs (18.4%). At that time, our members’ primary regions of work or study were the BC Lower Mainland or Fraser Valley (29%), Interior (10%), North (14%), Vancouver Island (41%), and Other or Outside BC (6%). The majority of our members (74.7%) are either studying or employed in the area of Indigenous health research.

### 2.5. ICCO-Specific Programs Offered

The BC NEIHR offers two ICCO-specific grants: (1) the Research Development Grant; and (2) the Knowledge-Sharing and Mobilization Grant. The Research Development Grant is offered annually and provides up to $10,000 CAD per project to assist ICCOs with community outreach, relationship building, and research-development activities. The Knowledge Sharing and Mobilization Grant is offered annually and provides up to $5000 CAD per project to support teams who have completed ICCO-led research and wish to share the findings of their research in culturally and contextually relevant ways.

### 2.6. Data Collection

From April 2020 to March 2023, the BC NEIHR created annual evaluation reports by collecting data on membership experiences in the form of surveys, interviews, and monthly journalling documentation from our Indigenous Health Research Facilitators and detailed minutes from over 800 h of meetings. Every year, we email a short survey to all ICCOs who have been funded and ask them to answer open-ended questions about how the funding has supported their self-determination and impacted their Indigenous health research journey. After each network event, we send out experience surveys to ICCOs to help us better understand the impact of our events and design future programming. Every month, each Indigenous Health Research Facilitator submits a report about their work, activities, and research relationship-building activities with ICCOs, including their own reflections on priorities, challenges, and success stories.

### 2.7. Analysis

We conducted a critical analysis of our annual evaluation reports and data to identify emerging best practices that will support the self-determination of ICCOs in health research. This included critically assessing successes, challenges, and lessons learned. In our analysis, we explored how Indigenous-led research networks can support the self-determination of ICCOs.

## 3. Results

Three best practices were identified to support the self-determination of ICCOs in health research: (1) creating capacity-bridging initiatives to overcome funding barriers; (2) building relational research relationships with ICCOs (“people on the ground”); and (3) establishing a network of partnerships and collaborations to support ICCO self-determination.

### 3.1. Best Practice 1: Creating Capacity-Bridging Initiatives to Overcome Funding Barriers

The BC NEIHR administers ICCO-specific grants to help create capacity for ICCOs to lead their own Indigenous health research in BC. Creating capacity-bridging initiatives to overcome the funding barriers currently experienced by ICCOs when trying to access or apply for research funding from mainstream funders emerged as a promising best practice. Over the past three years of operating the BC NEIHR, each ICCO project that was funded was place-based and autonomous, and the research was conducted by the community and for the community. Figure 1 below highlights the keywords used by ICCOs to describe their research within their 2021–2022 funding applications. The types of research activities led by ICCOs are fundamentally strength-based, culturally grounded, and self-determined.

#### 3.1.1. Supporting Community-Led Research in Funding Structures

To support the self-determination of ICCOs in health research, the BC NEIHR supports capacity bridging and mentorship that ensures ICCOs can participate in and control the research undertaken in their communities. To facilitate this, BC NEIHR ICCO-specific grants are held by ICCOs directly, and not by academic researchers at CIHR-recognized eligible host institutions. Administering funds to ICCOs directly helps to navigate and overcome the existing institutional eligibility barriers encountered by many ICCOs. Several ICCOs raised the negative experiences they had had when collaborating with universities on previous grant-funded projects; they encountered many hiccups and complications when working with the university’s accounting and finance teams, such as community members not receiving their compensation in a timely manner (if at all), or the university requesting personal information that was not necessary to process the compensation claims.

#### 3.1.2. Granting Support and Plain Language Applications

Part of creating capacity to overcome funding barriers involves ensuring that our grant application process is accessible and presented in plain language. The BC NEIHR regularly requests feedback from ICCO applicants about the process of applying for BC NEIHR funding, including whether the application form and instructions were easy to understand. While the majority of ICCOs raised no concerns, a few found the instructions difficult to understand. To help address the barriers around language that is accessible and relevant for ICCOs when applying for health research funding, we developed a series of materials to support ICCOs in their applications for BC NEIHR funding. These included a written workbook containing very clear examples/templates for each step of the application process and video-based nano-tutorials (i.e., videos that are 60–90 s long) for each step of the process. Furthermore, once an application for one of our funding opportunities is received, we follow an iterative, developmental review process whereby applicants submit a summary of their proposal for initial review and either (a) receive recommendations for improvement, revision, and immediate resubmission (iterative process), or (b) receive approval of their proposal for full review. For funded projects, a 10% holdback is released upon receipt at the end of a grant period of a final report which summarizes the processes and outcomes of the project.

#### 3.1.3. Enabling Development Projects to Be Competitive in Mainstream Funding

The creation of capacity-bridging initiatives has supported ICCOs and enabled them to lead the development of their own health research priorities and partnerships. ICCOs identified improving research partnerships as an important step toward ICCO leadership in research. Therefore, BC NEIHR ICCO-specific grants provide dedicated funds for community outreach and relationship building to lay the groundwork before research begins. This groundwork includes, but is not limited to, health research priority setting, partnership development, and application development. Allocated funds for research development are used to cover costs related to hiring research-development assistants, venue rental, travel, cash reimbursements (in a method acceptable to the individual or community being reimbursed) to honor Elders/Knowledge Holders and community participants, and culturally relevant promotional and gift items (e.g., cedar, blankets, food) and feasting. This represents critical support for increased submissions of ICCO-led applications to CIHR competitions and other funding agencies, and it improves the competitiveness of NEIHR-affiliated ICCOs and researchers. The use of BC NEIHR funding by ICCOs to conduct research development has resulted in several ICCOs securing funding from government agencies for future studies, though they are still ineligible to hold federal health research funds from the Tri-Agency in Canada.

### 3.2. Best Practice 2: Building Relational Research Relationships with ICCOs: “People on the Ground”

Building research relationships with ICCOs is an essential best practice. An approach that has been successful with the BC NEIHR is the implementation of the Indigenous Health Research Facilitator program. When developing the BC NEIHR, one of the means of support most strongly recommended by our ICCO stakeholders was “people on the ground” that can provide hands-on assistance to those who wish to prepare for and/or undertake health research. As a result, we have one Indigenous Health Research Facilitator for each health region in BC (i.e., Vancouver Island, Northern, Interior, Coastal Vancouver, and Fraser Valley), and they provide capacity-bridging support to ICCOs in their respective regions.

#### 3.2.1. Securing the Trust of ICCOs and Building Meaningful Research Relationships

Over the past three years, these Facilitators have taken the time to develop trusting, respectful, and meaningful research relationships with the ICCOs. The role of the Facilitators has been crucial in raising awareness of BC NEIHR health research opportunities and securing the trust of ICCOs, and this has led to the promotion of Indigenous-led health research in BC as well as ICCO success in major funding applications to external agencies. Once funded by the BC NEIHR, many of the ICCOs have remained connected and engaged within the network in meaningful ways, such as by collaboratively hosting network events in their traditional territories or entering into relationships with our partners.

#### 3.2.2. Network Values That Support Relational Research

Each of the Facilitators shares our values of self-determination, Indigenous knowledges and ethics, Indigenous and decolonizing methodologies, wholistic knowledge, and Indigenous cultural safety and equity. With this value system as our core and foundation, the Facilitators offer ICCOs wide-ranging, strength-based, and capacity-bridging activities, including creating and maintaining a supportive and culturally appropriate research environment that is welcoming to ICCOs and conducive to relationship-building, identifying and addressing regionally identified ICCO research priorities, helping to develop positive collaborative research partnerships, creating research-related resources for ICCOs interested in research development and knowledge sharing, and helping to support the development of community-based research ethics review processes. Overall, these activities have been instrumental in supporting ICCO self-determination and data sovereignty within the research environment.

#### 3.2.3. Accounting for Resources Needed for Relational Research

To build healthy relationships with ICCOs, the Facilitators have worked to make connections in various ways and to find the most beneficial and preferred communication styles for each ICCO. It has been a challenge to engage within cyberspace, i.e., to initiate and build relationships with ICCOs through email, social media, website messages and/or cold calling. This is likely due to multiple factors, including variable ICCO access to technology, geography (e.g., remote/rural communities), emergent flooding, and COVID-19 (e.g., community lockdowns, no events). Taking the time to meet someone “face-to-face”, either virtually via Zoom or by attending community health events, is an important aspect of relationality and building trust. We have heard from ICCO representatives that this relational communication has helped with anxiety about research and made them feel supported in the research process. The biggest challenges to developing and continuing relational research are the budget and resources, especially given the conventional timelines of research activities. For the BC NEIHR, reaching and connecting with ICCOs in the North has been particularly difficult given the cost of travel to and from the North. Future health research proposals and budgets must acknowledge that building research relationships with ICCOs requires in-person interactions and relationality, and that these are crucial to trusting, meaningful, respectful, and engaging relationships in the research environment as well as to ICCO self-determination.

### 3.3. Best Practice 3: Establishing a Network of Partnerships and Collaborations to Support ICCO Self-Determination

The BC NEIHR cannot support all the ICCO research needs in BC alone. Like-minded partners in this process are essential, not only in supporting ICCOs, but in pursuing other BC NEIHR research capacity-bridging goals. Given the limited budgets and resources for research programs today, culturally safe partnerships and relationships represent a critical best practice for supporting the self-determination of ICCOs in health research.

#### 3.3.1. Quality and Range of Partnerships Needed

The BC NEIHR represents a “network of networks” that is well-positioned to support ICCOs in leading their own health research by onboarding diverse partnerships and working together to create a more inclusive research environment. Our collaborative, multi-sectoral, multi-disciplinary partners include both Indigenous-led and non-Indigenous health authorities and organizations, Indigenous-led and community-based organizations, universities, mainstream funders, and other research bodies. The Michael Smith Health Research BC and BC SUPPORT Unit (funded by the BC SPOR—Strategy for Patient-Oriented Research) partnerships have been very important to the success of our programs. In 2021, leadership from the BC NEIHR and Michael Smith Health Research BC signed a Memorandum of Understanding (MOU) that guides our Indigenous-focused collaborative initiatives undertaken from 2021 to 2026 and describes the nature and principles of the relationship, including roles, responsibilities, and accountabilities.

#### 3.3.2. Partnership Activities and Contributions to ICCO-Led Health Research

To date, key partnership activities have included co-funding networking and capacity-bridging opportunities, communicating opportunities and initiatives through partner networks, and connecting ICCOs with partner networks to facilitate future research relationships. With the funds we receive from our partners, we can better support Indigenous research talent and offer all Facilitators full-time employment (rather than the half-time employment our budget permits), which means significantly more research-related support for the ICCOs in each region.

#### 3.3.3. Impact on Organizational Readiness for Change and Creating Safer Research Environments for ICCOs

All our partners strive to honor an explicit commitment to Indigenous health research leadership through equitable engagement and a service-oriented approach. Furthermore, through our partnerships, we contribute to organizational readiness for change and create more culturally safe environments for Indigenous health researchers and ICCOs who are working with or within non-Indigenous research organizations and structures. Through its partnership with the BC NEIHR, one mainstream health research funder accomplished transformational impacts at a structural and policy level, including a commitment to learning and practicing what Indigenous cultural safety looks like in action as a health research funder and realizing their vision of providing supportive research environments for Indigenous health research led by, and grounded in, Indigenous communities in BC.

## 4. Discussion

Many of the funding barriers and gaps that ICCOs continue to encounter stem from their ineligibility to hold and control health research funds awarded through mainstream funders. Helping ICCOs to become host institutions eligible to receive CIHR and other funds is an important step toward transforming the funding landscape. In supporting ICCOs through the BC NEIHR, we have seen that they possess all the capacities—to plan, to organize, to operationalize—to lead. They simply require support and opportunities to apply those capacities in the context of research and within what Indigenous scholar Willie Ermine refers to as the Ethical Space, where Indigenous worldviews, knowledge systems, and practices are valued, acknowledged, and used to ground research [31]. Supporting self-determination by first addressing the funding barriers and prioritizing Indigenous ways of knowing is crucial to building greater research capacity within ICCOs. Through health research networks such as the BC NEIHR, ICCOs can increase their competitive and awarded funding opportunities. However, there are challenges and limitations that may prevent ICCOs from holding their own research funds. Certainly, there are varying research resources and capacities among ICCOs. In BC, there are over 300 ICCOs, and we have seen that larger, more urban ICCOs are often in a position to conduct large-scale quantitative studies, while smaller, more isolated ICCOs are disproportionately negatively affected by fewer staff and resources. The latter are frequently the least equipped to carry out research to back up their needs and at best can usually only lead small-scale qualitative summaries. This is an issue that needs to be addressed because it is inequitable to only support the self-determination of ICCOs with high research capacity. Another challenge is that inequities are more common in some regions, such as the North in BC. Having strategic interventions that are regionally based would be an important consideration for large provincial health research networks.

Key to ICCO-led and self-determining research development and knowledge sharing are relational approaches and relationship building. ICCOs continue to be fearful and mistrustful of research due to countless generations of misrepresentation and exploitation in which they have been the subjects of academic, “scientific” studies conducted by non-Indigenous people [19,20,27,29,32,33,34]. Prioritizing relationship building with our Facilitators as the first step has been part of the success of the BC NEIHR because it alleviates some of the fear and mistrust of research. Once a culturally safe and capacity-bridging environment that encourages Indigenous ways of knowing and being is established, we can then support ICCO self-determination and introduce them to other research partners with whom we are working, such as the health authorities or BC SUPPORT Units. Positive research experiences and relationships have the power to connect individuals, groups, organizations, and institutions and transform the health research environment into one that is more inclusive; it has a positive compounding and spiraling effect [35]. In our experience, health research networks have proven to be a way to build beneficial research experiences and relationships. The challenge of culturally relevant and appropriate research relationship building with ICCOs is that research budgets and timelines often do not adequately meet the requirements for building such trusting and meaningful relationships. Another limitation is sustainability of research programs. The funding of health research networks such as the NEIHR program lasts for only one term, with no guarantee of renewal. This has important implications because the end of a program can be hurtful to ICCO-based research relationships and potentially reinstate mistrust of research.

Western epistemologies, ontologies, and methodologies have generated core research institutions and systems that have determined how the research enterprise is structured. That is, the core environments, systems, and institutions of research determine what is seen and what is ignored, what is valued and what is rejected, what is protected and what is neglected. The BC NEIHR aims to catalyze a shift in the present research environments, systems, and institutions through our strong and collaborative partnerships. In our partnerships, we have found a balance that both works with and pushes back against these structures so that ICCOs can engage in health research to their full potential. We cannot fully support ICCO self-determination in research by working in silos. We must work together to create a healthier, safer, and more inclusive research environment that respects and prioritizes Indigenous ways of knowing and being.

## 5. Conclusions

Indigenous-led health research networks that support self-determination and prioritize Indigenous ways of knowing and being are crucial to both capacity-bridging and reconciliation efforts in Canada. We identified three best practices to support the self-determination of ICCOs in health research that could be used by other networks or organizations to further advance Indigenous-led, culturally grounded, and self-determining research. The BC NEIHR has learned that through meaningful engagement with diverse partners, we can collectively work to co-create opportunities for ethical and impactful partnerships between ICCOs and health researchers, scholars, research administrators, policy and decision makers, and other parties with an interest in respectful Indigenous health research.

Communities are calling for ICCO leadership in research so that knowledge about Indigenous health will no longer be possessed exclusively by mainstream institutions, creating the potential for further marginalization and stigmatization of Indigenous peoples. Our focus on local Indigenous knowledge systems, cultures, and contexts provides an excellent foundation for balanced health research that supports ICCO self-determination. Through networks such as the BC NEIHR, there is the potential to change the landscape of health research.

## Figures and Tables

**Figure 1 ijerph-20-06523-f001:**
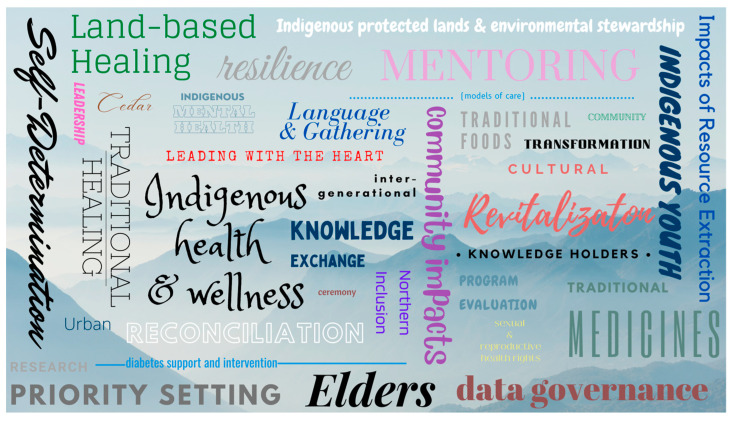
The keywords used by ICCOs in their 2021–2022 BC NEIHR funding applications.

## Data Availability

Data sharing not applicable.

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
