# Peer review of "Best Practices to Support the Self-Determination of Indigenous Communities, Collectives, and Organizations in Health Research through a Provincial Health Research Network Environment in British Columbia, Canada"

_ijerph, 2023, doi:10.3390/ijerph20156523_

Round 1
Reviewer 1 Report
The article focuses on analyzing a very interesting topic. In this sense I would like to congratulate the authors. However, as it is written, I currently consider that the article should not be published. Fundamental reasons are as follows:
1. The study is descriptive in nature, and does not provide elements of interest to the scientific community. The set of institutions and programs are of interest, but they are not adequately problematized. It is necessary to analyze pros and cons. I wonder if a strengths and weaknesses analysis would have helped. A hermeneutic or phenomenological analysis could also have been done. The important thing would be to go deeper and provide new and profound knowledge.
2. Many statements are made that are not confirmed with data or verifiable information. For example, there is talk of paternalism, it is also stated that "The role of the IHRFs has been crucial in raising awareness of the BC NEIHR health research opportunities among ICCOs and securing the trust of ICCOs". The problem is that this and other claims are not well substantiated. In fact, the article shows no data to substantiate the information in any way.
3. It is necessary to introduce a section or, at least, a methodological section explaining how the analysis was carried out, what type of analysis and the time frame.
4. Authors mention self-determination and ethics. I suggest that only one of the two elements be analyzed. I say this because, humbly, I think it would be necessary to develop the concepts further, to go deeper into existing theories, to contrast these theories with the article, etc.
5. Finally, this paper is extremely unclear and its reading is difficult. Authors use a multitude of acronyms. It is essential to simplify the work to make it understandable.
Author Response
- The paper is descriptive in nature and provides a narrative or story-based analysis of the network. Based on the dearth of literature on previous networks (e.g., the NEAHR program that was de-funded in 2011), this information on lessons learned is valuable and a contribution to the scientific community as well as funders who are interested in developing Indigenous health research networks. We have more adequately problematized the network with added strengths and weaknesses, particularly in the form of ongoing challenges. We added a deeper analysis and offered new knowledge.
- We removed statements that were not confirmed by data or verifiable information. We added more data from funded ICCOs about their experience within the network to substantiate our claims. For example, we offer information from our annual evaluation reports and detail how we analysed that information. While this article is not based on original research, this network and the narrative we provide is based on and shows the importance of culturally appropriate and community-driven processes.
- We offer more information on our methodology and methods. We explain how the analysis was carried out and the time frame.
- We narrowed the focus to how Indigenous health research networks support ICCO self-determination.
- We removed unnecessary acronyms to improve clarity and make it more understandable. We completely changed the structure/format of the paper to make it flow and easier to follow.
Reviewer 2 Report
Abstract:
The abstract lacks specific examples or evidence to support the claim that Indigenous-led research networks are transforming the funding landscape in Canada. It would benefit from providing concrete data or case studies in one line to demonstrate the impact of these networks.
The abstract does not mention any limitations or critiques of the BC NEIHR or Indigenous-led research networks in general. It would be valuable to acknowledge and address potential challenges or criticisms in one line to provide a more balanced perspective.
The abstract does not clearly outline the methodology or approach used in the paper, making it difficult to evaluate the rigor of the research or the reliability of the lessons learned. Including information on the research methods employed would enhance the abstract's clarity and credibility.
Introduction:
The introduction lacks specific references to support the claims made, such as the limitations of the current health research funding landscape or the need for transformations in Indigenous health research. Including specific citations or studies would enhance the credibility of the statements.
The introduction does not clearly articulate the gap or problem that the BC NEIHR aims to address. It would benefit from explicitly stating the research gap or the specific challenges faced by Indigenous communities in the current funding landscape.
The introduction does not provide a clear research objective or research question that the paper aims to address. It would be helpful to clearly state the purpose of the study, such as identifying the key successes, challenges, and lessons learned from operating the BC NEIHR.
Lack of critical analysis: The manuscript provides a historical overview of the NEAHR program without critically analyzing its limitations or shortcomings, such as the potential challenges faced during its implementation or any criticisms from Indigenous communities or researchers.
Inadequate discussion of the termination of NEAHR Centres: The manuscript briefly mentions that the NEAHR Centres were terminated due to conservative government cuts, but it does not delve into the specific detrimental impacts on community-led Indigenous health research and knowledge sharing. A more comprehensive discussion of these impacts would provide a better understanding of the challenges faced by the program.
Limited exploration of ethical considerations: Although the manuscript briefly mentions the principles of Ownership, Control, Access, and Possession (OCAP®) and other ethical guidelines, it does not thoroughly examine the ethical tensions and challenges associated with conducting research with Indigenous communities. A more in-depth exploration of ethical considerations would enhance the discussion on supporting Indigenous governance and self-determination in health research.
The manuscript does not discuss the specific outcomes or impact of the funded projects, making it difficult to assess the overall success or effectiveness of the BC NEIHR programs.
The manuscript does not address the potential sustainability or long-term impact of the partnerships and collaborations established by the BC NEIHR, leaving unanswered questions about the continued support for Indigenous health research beyond the scope of the programs.
Author Response
Abstract:
- We have changed the wording around Indigenous health research networks transforming the funding landscape to Indigenous health research networks being a response to the current landscape and then discuss in more detail the barriers to the current landscape. There is a dearth of literature on previous or other Indigenous health research networks in Canada. Because of that there is little data or case studies to draw on. The BC NEIHR lessons learned and impact can fill this gap in the literature and be an important contribution to the scientific community as well as funders who are interested in developing Indigenous health research networks.
- We offer more limitations and critiques of the BC NEIHR, such as the geographic and cultural diversity of BC ICCOs, limited funds for relationship-building and outreach (essential to building trusting and meaningful relationships with ICCOs), institutional barriers for ICCOs, etc.
- We outline our methodology and methods in more detail as well as better explain how we conducted the analysis. This way readers can better evaluate the reliability of our lessons learned.
Introduction:
- We removed claims that did not have references and added more citations for the claims that we do make.
- We more clearly state the gap/problem that the network is addressing. We explicitly focus on how Indigenous health research networks can support ICCO self-determination, which is the research gap faced by ICCOs in the current funding landscape.
- We provide a clearer objective/purpose that the paper aims to address.
- We offer a more critical analysis. Because of the lack of literature on Indigenous health research networks in Canada, we cannot provide a substantial historical overview of the previous NEAHRs (e.g., its shortcomings and limitations, challenges faced during information, or its termination). But we did completely change the structure/format of this paper to offer a more comprehensive discussion of the lessons learned and impact of the BC NEIHR, which fill the gap in the literature.
- We offer a thoroughly discussion of ethics and ethical tensions in section 1.1. The Need for Indigenous Leadership, Self-Determination, and Governance in Health Research, but restructured the paper to be a more narrow analysis of how Indigenous health research networks support ICCO self-determination.
- We provided more information on the specific outcomes and impact on funded ICCO projects. We offer more insight into organizational change and long-term impact with our partners.
Round 2
Reviewer 1 Report
The paper has been substantially improved. In my opinion it is now much clearer and the ideas, the process carried out, the results and the conclusions are well expressed. I appreciate the great effort made by the authors.
Reviewer 2 Report
In better form now.